

# Tea cultivation: facilitating soil organic carbon accumulation and altering soil bacterial community—Leishan County, Guizhou Province, Southwest China

Yingge Shu, Shan Xie, Hong Fan, Chun Duan, Yuansheng Liu and Zuyong Chen

College of Agronomy, Guizhou University, Guiyang, Guizhou, China

Corresponding author
Zuyong Chen,
qingfeng340@126.com

## ABSTRACT

**Background:** *Camellia sinensis* is an important cash crop in southwestern China, with soil organic carbon playing a vital role in soil fertility, and microorganisms contributing significantly to nutrient cycling, thus both of them influencing tea tree growth and development. However, existing studies primarily focus on soil organic carbon, neglecting carbon fractions, and the relationship between soil organic carbon fractions and microbial communities is unclear. Consequently, this study aims to clarify the impact of different tea planting durations on soil organic carbon fractions and microbial communities and identify the main factors influencing microbial communities. It provides a theoretical basis for soil quality evaluation in the study area and scientific guidance for tea plantation management, thus fostering the region's economic sustainability.

**Methods:** This study selected tea plantations with different tea planting durations of 3–5 years (Y5), 12–16 years (Y15), 18–22 years (Y20), 40–42 years (Y40), and 48–50 years (Y50), as research subjects and adjacent uncultivated forest without a history of tea planting (CK) served as controls. Soil organic carbon (SOC), particulate organic carbon (POC), easily oxidizable organic carbon (EOC), dissolved organic carbon (DOC), microbial biomass carbon (MBC), and bacterial diversity were measured in the 0–20 cm and 20–40 cm soil layers, respectively.

**Results:** Compared to the adjacent uncultivated forest (CK), the soil organic carbon (SOC), easily oxidizable carbon (EOC), particulate organic carbon (POC), and dissolved organic carbon (DOC) contents in a 40-year tea plantation significantly increased. Nonetheless, the microbial biomass carbon (MBC) content notably decreased. POC/SOC ratios rose with prolonged planting, signifying enhanced conversion of organic carbon into particulate forms. Bacterial community diversity peaked at 15 years and declined by 40 years post-planting and after tea planting dominated by *Acidobacteriota*, *Chloroflexi*, *Proteobacteria*, and *Actinobacteriota* in the tea garden. FAPROTAX analysis highlighted aerobic and anaerobic chemoheterotrophy, cellulolysis, and nitrogen fixation as key bacterial functions. POC and MBC significantly influenced bacterial community structure. In conclusion, tea plantation soil exhibited the highest organic carbon content at 40 years of tea planting, indicating strong carbon accumulation capacity. However, soil acidification in the tea plantation may affect changes in organic carbon and bacterial community. Therefore, in the tea planting process, it is necessary to improve the

management system of tea plantations to ensure the maintenance of a good ecological environment in the tea plantation soil, thus achieving sustainable development of the tea industry in the region.

## INTRODUCTION

*Camellia sinensis* is a perennial evergreen woody economic crop originating in Guizhou Plateau in southwestern China. The global tea market is vast and plays a crucial role in driving exports and economic growth in numerous developing countries (*Han, Kemmitt & Brookes, 2007*; *Zhao et al., 2022*). Soil organic carbon is a crucial component of soil fertility, influencing the physicochemical and biological characteristics of soils, and their evolution is affected by their quality and quantity. It plays an important role in maintaining and coordinating the supply and storage of soil nutrients, influencing the types and activities of soil enzymes, controlling the primary and secondary production of plants, as well as the cycling of matter and energy flow in ecosystems (*Priess, de Koning & Veldkamp, 2001*; *Gaiser & Stahr, 2013*). Although dissolved organic carbon, microbial biomass carbon, and other active carbon components constitute a small proportion of organic carbon, they exhibit high sensitivity to changes in soil organic carbon transformation rates and quality under different agricultural management practices. Therefore, they are the most commonly used carbon components in soil quality assessment (*Li et al., 2021*).

The high vegetation cover of tea plantations and the high biomass of tea trees and photosynthesis increase the carbon sequestration capacity of tea plantation vegetation and achieve soil carbon accumulation through the input of vegetation carbon (*Wang et al., 2023b*). Organic litter (such as dead leaves, branches, and pruning residues) in tea gardens provides a significant source of soil organic carbon. During their decomposition process, the comparatively resistant chemical components within these litter materials, alongside the substantial release of soluble organic carbon from decomposed pruning leaves, collectively form an integral part of soil organic matter (*Wallenstein et al., 2013*; *Zhou et al., 2024*). From the perspective of countries or regions, research conducted in areas such as Fuan City in China, Xishuangbanna in Yunnan, and the western hills of Sichuan has shown that higher moisture content, fertilization, and input of organic matter increase soil organic carbon (*Wang et al., 2018*; *Di et al., 2020*; *Wang, Yao & Ye, 2020*). In regions of Kenya, the favorable climate provides an excellent environment for the survival of tea trees, and the high altitude and low temperatures decelerate the decomposition of organic matter, similarly leading to high soil organic carbon content (*Chiti et al., 2018*). However, studies in areas such as the Turkish region, India, and Anxi in Fujian, China, have shown a decline in organic carbon due to exposure to anthropogenic activities, water scouring, and intensive farming (*Yuksek et al., 2009*; *Kalita, Das & Nath, 2016*; *Shui, Wu & Fu, 2022*). It can be seen that there are differences in the research on soil organic carbon in tea plantations among different regions, primarily focusing on organic carbon, with a lack of

research on soil carbon fractions in tea plantations. At the same time, before our study, no one had studied the soil organic carbon in the tea planting area of Leishan County, Guizhou Province, in the southwestern region of China. Therefore, studying the characteristics of soil organic carbon and its components in tea plantations of different planting years is of great significance for the carbon balance of tea plantation soil in this research area and for soil quality assessment. However, the storage and transformation of organic carbon are affected by various factors. Currently, research also focuses on the effects of cultivation and fertilization on organic carbon. Practices such as reduced tillage and no-till farming can increase the return of vegetation biomass, reduce soil layer disturbance, and slow down the decomposition of soil organic carbon, making them more conducive to increasing soil organic carbon reserves and enhancing soil carbon pool levels compared to traditional tillage (*Badagliacca et al., 2018*; *Jat et al., 2019*). Therefore, studying soil organic carbon pools is a significant approach to exploring soil carbon sequestration capacity and further investigating the underlying mechanisms of how soil organic carbon affects soil quality.

Soil microorganisms participate in about 90% of soil reaction processes and are sensitive to changes in soil chemical properties. They play an important role in soil organic matter decomposition and mineral nutrient cycling, which is significant for evaluating soil quality (*Philippot et al., 2024*). With changes in the years of tea cultivation, not only does the tea plantation soil environment undergo significant changes, but the fertility of the tea plantation soil also shows a declining trend (*Zhang et al., 2021b*). Soil microorganisms have the advantages of nitrogen fixation, potassium and phosphorus solubilization, decomposition of organic matter, and enhancement of moisture retention, which help the growth of tea buds, metabolism of the tea plant, production of unique tea aroma substances, reduction of pests and diseases, and significantly improve the yield and quality of tea leaves (*Zhou & Chen, 2014*). In the aspect of tea plantation soil microorganism research, previous studies have made some progress, *Li et al. (2019b)* showed that the abundance of soil microorganisms in the inter-root zone of tea trees was significantly affected by the mode of fertilization, and the carbon and nitrogen content of inter-root zone soil microbial biomass and enzyme activities were significantly increased compared with no fertilization. *Yuan et al. (2015)* indicated that polyphenolic substances in the litter of tea trees gradually accumulate in the soil and that certain inhibitory compounds generated during their degradation process affect the structure of soil microbial communities. *Yang et al. (2011)* found that the number of years of tea planting had a direct effect on the soil microbial population, which showed a trend of first increasing and then decreasing with the increase in the number of years of tea planting. Research has also reported that with the increase in years of tea cultivation, there are changes in the structure of the tea plant rhizosphere soil microbial community, primarily characterized by decreased diversity. Within a certain range of years, the longer the cultivation period, the lower the diversity index. Additionally, there is an increase in microbial species with strong adaptability and metabolic activity. However, when the tea cultivation time reaches around 80 years, the diversity of soil microorganisms shows a trend of initially increasing and then decreasing, reaching its maximum at around 40 or 50 years (*Xue, Yao & Huang, 2006*; *Lin*

*et al., 2013*). It is obvious that there are differences in the current research on soil microorganisms in tea plantations. Therefore, studying the diversity of soil microbial communities in tea plantations under different years of cultivation is beneficial. By gaining insight into the sensitivity feedback of soil microorganisms to soil environmental factors, we can promptly adjust fertility management practices in tea plantations and provide theoretical support for their sustainable development.

Therefore, based on previous research, to provide suitable management practices for tea plantations in this region, we hypothesize that (1) long-term tea cultivation benefits the transformation and accumulation of soil organic carbon in tea plantations and (2) different tea cultivation times result in differences in soil bacterial community structure. Thus, the objectives of this study are: (1) to investigate the changes in soil organic carbon and its active components in tea plantations with different years of tea cultivation in Leishan County, Guizhou Province, Southwest China; (2) to analyze the differences in soil bacterial communities under different years of tea cultivation; and (3) to clarify the relationship between soil organic carbon and its active components and soil bacterial communities after tea cultivation. These research findings can provide scientific evidence and contribute to the sustainable development of soil in this region.

## MATERIALS AND METHODS

### Regional study overview

The study region which is located in Leishan County, Guizhou Province (26°02′–26°34′N, 107°55′–108°22′E), belongs to the slope zone of the transition between the Yunnan-Guizhou Plateau and the Xianggui Hilly Basin, with an average elevation of 1,329.4 m. The area has a humid climate in the mid-subtropical monsoon and a mild climate, featuring an average annual precipitation of 1,375 mm, an average annual air temperature of 15.6 °C, and a year-round sunshine duration of 1,225 h, with abundant sunshine and rainfall. The locations of the sampling points are shown in Fig. 1.

### Design of the experiment

In late July 2023, based on comprehensive field surveys conducted in the study area, considering topography, elevation, parent material, soil type, fertilization and management practices, as well as the years of tea cultivation, a total of five tea garden soils were selected for study within the altitude range of 1,100 to 1,250 m. These selected tea gardens exhibited relatively consistent environmental factors and fertilization management practices but varied in the years of tea cultivation (ranging from 3 to 5 years, 12 to 16 years, 18 to 22 years, 40 to 42 years, and 48 to 50 years). Additionally, adjacent uncultivated forest, devoid of tea cultivation history, was chosen as a control (representing 0 years of tea cultivation). Due to the extended cultivation periods in some tea gardens, the exact years of tea cultivation were estimated based on the recollection of elderly villagers in the area, resulting in approximate timeframes rather than precise durations. Furthermore, for cartography and analysis, the various years of tea cultivation mentioned above were denoted sequentially as CK, Y5, Y15, Y20, Y40, and Y50, respectively. The basic

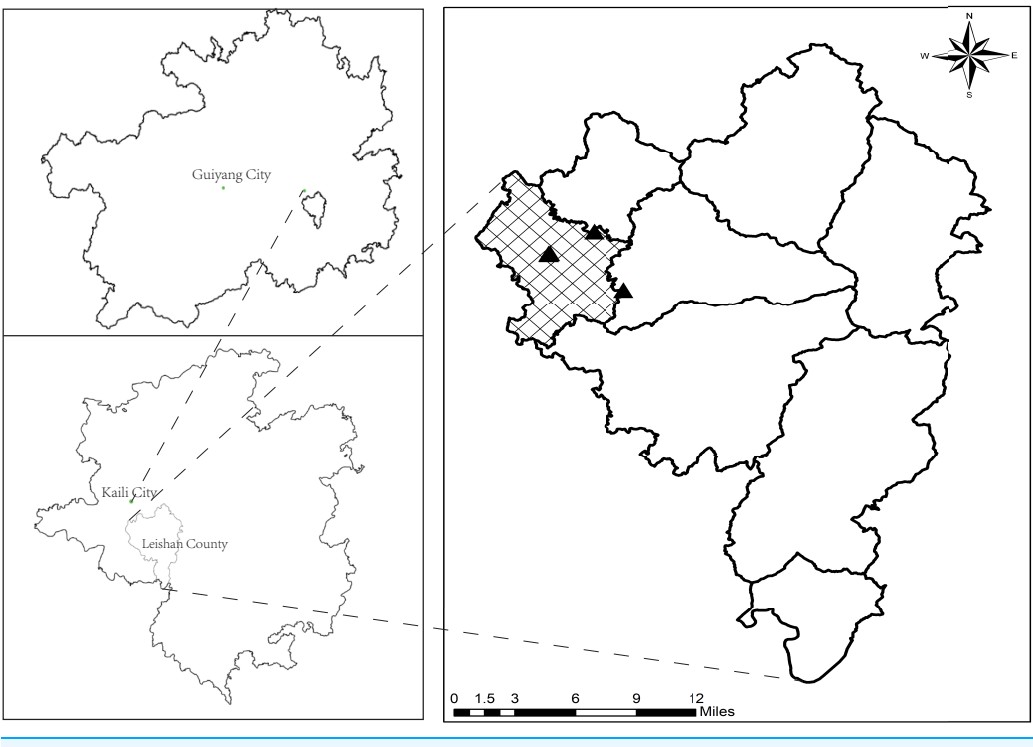

**Figure 1 The location of sampling points.**

physico-chemical properties of tea plantation soils in the study area for each year of tea planting are shown in Table 1.

## Soil sample collection

In tea gardens of different cultivation durations, typical sampling zones were established at the upper, middle, and lower sections of tea gardens on the same slope and aspect. Within each sampling zone, three mixing sampling points were designated, alongside three corresponding control points, resulting in a total of 60 mixing sampling profiles excavated (Number of profiles: 3 * 3 * 5 + 5 * 3 = 60). At each mixing sampling point removing the surface litter, a simple soil profile was dug at the drip line along the outer perimeter of the plant canopy. Then we collected soil from 0–20 and 20–40 cm soil horizons, respectively, and mixed soil samples from three replicates of the same type of mixing point within the same sampling area into one mixed sample. Soil samples were sealed in sampling bags, refrigerated, and transported back to the laboratory. One portion of the soil was air-dried, ground, and passed through a nylon sieve for further analysis, while another portion was stored in a low-temperature refrigerator for determination of soil microbial biomass carbon and nitrogen, as well as bacterial community diversity. Since the top soil layer of tea garden soil is more susceptible to the influence of tillage management and soil litters, the bacterial community changes in the 0–20 cm soil layer were mainly investigated in this study.

**Table 1 The basic physiochemical properties of tea plantation soils with different planting ages.**

| Soil layer | Years | pH | TN (g·kg⁻¹) | AN (mg·kg⁻¹) | AP (mg·kg⁻¹) | AK (mg·kg⁻¹) |
|---|---|---|---|---|---|---|
| 0–20 cm | 0 | 4.58 ± 0.01Aa | 1.65 ± 0.12Ad | 132.23 ± 29.73Ac | 1.14 ± 0.53Ad | 76.00 ± 1.73Ac |
| | 5 | 4.50 ± 0.18Aab | 2.48 ± 0.54Acd | 173.30 ± 41.99Abc | 3.87 ± 0.29Ad | 79.33 ± 29.14Ac |
| | 15 | 4.26 ± 0.08Bbc | 2.96 ± 0.11Ac | 227.10 ± 15.84Ab | 7.17 ± 1.71Ad | 94.00 ± 24.56Ac |
| | 20 | 4.11 ± 0.18Ac | 3.29 ± 1.24Abc | 222.99 ± 42.75Ab | 24.29 ± 6.76Ac | 124.67 ± 11.06Ac |
| | 40 | 3.66 ± 0.33Ad | 5.04 ± 0.95Aa | 348.88 ± 55.85Aa | 69.41 ± 11.95Aa | 411.33 ± 63.63Aa |
| | 50 | 3.25 ± 0.05Be | 4.43 ± 0.48Aab | 334.28 ± 47.32Aa | 55.34 ± 10.45Ab | 196.00 ± 54.95Ab |
| 20–40 cm | 0 | 4.63 ± 0.03Aa | 0.91 ± 0.22Bb | 57.49 ± 3.70B6c | 0.34 ± 0.20Bd | 55.33 ± 29.94Ac |
| | 5 | 4.50 ± 0.14Aa | 0.70 ± 0.21Bb | 98.19 ± 106.18Bc | 1.33 ± 0.50Bd | 42.00 ± 19.70Ac |
| | 15 | 4.47 ± 0.06Aa | 2.01 ± 0.51Ba | 118.21 ± 121.91Ab | 1.20 ± 0.46Bd | 37.67 ± 15.14Bc |
| | 20 | 4.43 ± 0.11Aab | 2.38 ± 0.52Aa | 123.24 ± 128.89Bab | 11.51 ± 4.75Ac | 68.83 ± 5.92Bbc |
| | 40 | 4.07 ± 0.47Ab | 2.15 ± 0.13Ba | 189.62 ± 196.00Ba | 30.28 ± 6.84Ba | 259.67 ± 32.35Ba |
| | 50 | 3.65 ± 0.13Ac | 2.28 ± 0.47Ba | 180.08 ± 185.09Ba | 21.44 ± 0.29Ab | 112.00 ± 54.06Ab |

**Note:**
Significant differences between different soil layers of the same tea age are indicated by different capital letters ($p < 0.05$). Significant differences between different periods of the same soil depth are indicated by different lowercase letters ($p < 0.05$).

## Soil physiochemical analyses

Soil pH value measured by pH meter (water-soil ratio 2.5:1) (Remco PHS-3C model); available nitrogen decomposition by alkaline diffusion; available phosphorus by $NH_4F$ leaching colorimetric method (METASH UV-5500 UV-Vis Spectrophotometer); available potassium $NH_4OAc$ by leaching-flame photometric method (Jingruo FP6420 flaming photometer); soil total nitrogen by semi-micro Kjeldahl method (Peiou SKD-200 Kjeldahl analyzer). Soil organic carbon (SOC) by $K_2Cr_2O_7$-$H_2SO_4$ external heating method (Guohua HH-S Oil Bath); easily oxidizable organic carbon (EOC) was determined by oxidation with 333 mmol/L $KMnO_4$ (METASH UV-5500 UV-Vis Spectrophotometer); the particulate organic carbon (POC) was extracted by separating it with sodium hexametaphosphate solution through a 0.053 mm sieve, then dried and externally heated with potassium dichromate-sulphuric acid (Guohua HH-S Oil Bath); dissolved organic carbon (DOC) was determined by 0.5 mol/L $K_2SO_4$ leaching, and microbial biomass carbon (MBC) was determined by chloroform fumigation-0.5 mol/L $K_2SO_4$ leaching, and the leachate was filtered through a 0.45 µm filter membrane and determined by a total organic carbon analyzer (Vario, TOC, Germany).

## Soil bacterial diversity
### DNA extraction and PCR amplification

Soil samples were sent to Shanghai Biozeron company for sequencing using the Illumina NovaSeq 2500 high-throughput sequencing platform. Microbial DNA was extracted from soil samples using the E.Z.N.A.® Soil DNA kit (Omega Bio-Tek, Norcross, GA, USA) following the manufacturer's protocols. PCR amplification of the V3–V4 region of the bacterial 16s RNA gene was carried out (95 °C for 2 min, followed by 25 cycles at 95 °C for 30 s, 55 °C for 30 s, and 72 °C for 30 s, with a final extension at 72 °C for 5 min) using primers 341F (5′-CCTAYGGGRBGCASCAG-3′) and 806R (5′-GGACTACNNGGGTA

TCTAAT-3′), each sample labeled with a unique eight-base barcode. Finally, the purification was performed according to the instructions of the AxyPrep DNA Gel Extraction Kit (Axygen Biosciences, Union City, CA, U.S.).

## Alpha and beta diversity analyses

Rarefaction analysis was conducted using Mothur v.1.21.1 (*Schloss et al., 2009*), while beta diversity was assessed through UniFrac (*Lozupone et al., 2011*) to compare the results of the non-metric multidimensional scaling analysis (NMDS). It simplifies research samples to low-dimensional space for analysis, preserving original relationships. It reflects species information as points in multidimensional space, with differences shown by point distances. Stress <0.2000 indicates analysis reliability (*Dexter, Rollwagen-Bollens & Bollens, 2018*).

### Functional prediction

Predicting biochemical cycling functions of endophytic bacterial communities using FAPROTAX software (1.2.1) (*Louca, Parfrey & Doebeli, 2016*).

### Data processing and analysis

Excel 2010 and Data Processing System 7.05 (DPS 7.05) were used for data analysis and one-way ANOVA (LSD method); Origin 2024 was used for mapping; and Canoco 5.0 was used to perform redundancy analysis (RDA) of soil environment and microbial communities.

## RESULTS

## Characteristics of soil organic carbon and its components with different tea planting duration

The characteristics of soil organic carbon content changes in different tea planting years in the study area are shown in Fig. 2, and tea planting promoted the accumulation of soil organic carbon. In the 0–20 cm soil layer, the changes in organic carbon and component contents from CK to Y50 are shown in Fig. 2A. The average SOC contents ranged from 17.26 to 60.91 g·kg$^{-1}$, with a maximum value at Y40. Among all tea planting years, Y20, Y40, and Y50 were 92.58%, 252.90% and 229.20% higher than CK, respectively ($p < 0.05$), and the organic carbon contents of Y5 and Y15 were also higher than CK, but there was no notable difference. In addition, the soil organic carbon content was significantly higher ($p < 0.05$) in the long time of tea planting (Y40 and Y50) than in the early stage of tea planting (Y5 and Y15). The average EOC content ranged from 3.79 to 12.97 g·kg$^{-1}$, and the EOC content in the 40th and 50th year of planting was markedly higher than in the other years of planting ($p < 0.05$), and compared to CK and Y5, the increase in Y50 was respectively 242.22% and 130.78%. The average POC content ranged from 8.09 to 43.21 g·kg$^{-1}$, and the POC content in 20, 40, and 50 years of tea planting was significantly different from that of CK and Y5 ($p < 0.05$), with the largest difference in 40 years, which was significantly more than 5.34 and 3.66 times higher than CK and Y5, separately. The average DOC content ranged from 123.48 to 214.95 mg·kg$^{-1}$, and compared with CK, the DOC content of Y5, Y15, and Y20 decreased by 30.90%, 12.42%, and 12.81%, respectively,

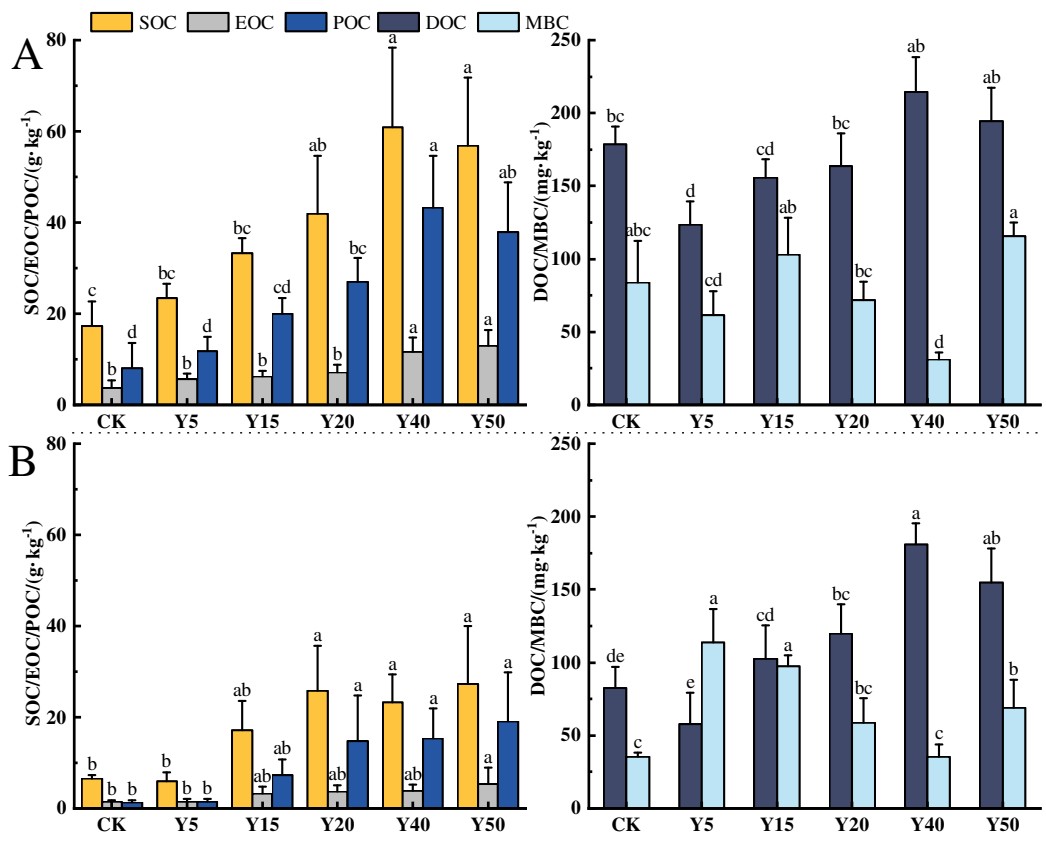

**Figure 2 Characteristics of SOC fractions in different tea planting period.** (A) and (B) represent the 0–20 cm and the 20–40 cm soil layer, respectively. Significant differences between different periods are indicated by different lowercase letters ($p < 0.05$).

with a significant decrease in Y5 ($p < 0.05$). The average content of MBC ranged from 31.28 to 115.73 mg·kg$^{-1}$, with the lowest content in Y40, which was significantly lower than the other tea planting years. Compared with CK, Y15 and Y50 were upward changes but not significantly different, while Y40 was significantly decreased by 62.79% ($p < 0.05$).

In the 20~40 cm depth of the soil, the changes in organic carbon and its components from CK to Y50 for each tea cultivation year are shown in Fig. 2B. The average SOC content ranges from 5.91 to 27.36 g·kg$^{-1}$, with the lowest content in Y5. After tea cultivation, the SOC in the Y5 year decreased by 9.22% compared to CK, but there was no significant difference. Subsequently, with the increase of tea cultivation years, SOC content gradually increased, reaching an average of 25.76 g·kg$^{-1}$ at Y20, which was 3.36 times higher than Y5 ($p < 0.05$). The average EOC content ranges from 1.47 to 5.44 g·kg$^{-1}$. EOC content in tea garden soil gradually accumulates between 5 and 50 years of tea cultivation. There are significant differences in EOC content between Y50 and CK and Y5, increasing by 270.07% and 257.89%, respectively compared to CK and Y5 ($p < 0.05$). The average POC content ranges from 1.41 to 19.15 g·kg$^{-1}$, gradually increasing with the extension of tea cultivation years. There are significant differences in POC content between Y20, Y40, Y50, and CK, with Y20, Y40, and Y50 showing significant increases of 17.74, 13.90, and 13.31 g·kg$^{-1}$, respectively, compared to CK ($p < 0.05$). The average DOC content ranges

from 57.62 to 180.93 mg·kg$^{-1}$, with the lowest content in Y5, showing a decrease of 30.06% compared to CK. The average MBC content ranges from 35.22 to 113.72 mg·kg$^{-1}$, with the lowest MBC content in Y40 and the highest in Y5, with a difference of 2.23 times between them.

## Characteristics of soil organic carbon fractions as a proportion of organic carbon with different tea planting duration

The proportions of each organic carbon fraction to organic carbon in the soils of tea plantations of different tea planting years are shown in Table 2 below. Significant variations in these ratios across different planting years are evident. In the 0–20 cm and 20–40 cm soil layers, the proportion of EOC/SOC ranged from 17.22% to 23.91% and 14.98% to 26.00%, with an average of 20.65% and 19.72%; the proportion of POC/SOC ranged from 44.42% to 71.31% and 21.46% to 68.09%, with an average of 59.83% and 45.53%; the proportion of DOC/SOC was from 44.42% to 71.31% and 21.46% to 68.09%, respectively. A total of 59.83% and 45.53%; the proportion of DOC/SOC ranged from 0.36% to 1.09% and 0.50% to 1.26%, with an average proportion of 0.53% and 0.79%; and the proportion of MBC/SOC ranged from 0.05% to 0.54% and 0.15% to 2.11%, with an average proportion of 0.26% and 0.66%. From this, it can be concluded that the order of high and low organic carbon in the proportion of each carbon fraction is POC > EOC > DOC > MBC, which is mainly dominated by POC, followed by EOC, and MBC occupies the smallest proportion. Among them, relative to the organic carbon between different soil layers, EOC and POC accounted for a higher proportion of organic carbon in the 0–20 cm soil layer, while DOC and MBC accounted for a higher proportion of organic carbon in the 20–40 cm depth of the soil.

In the 0–20 cm depth of the soil, the proportion of EOC/SOC had the highest proportion at Y5, which was significantly higher than Y5 and Y15 by 4.96 and 6.69 percentage points, respectively ($p < 0.05$). The proportion of POC/SOC had the highest proportion at Y40, which differed significantly from CK and Y5, and was significantly higher than CK and Y5 by 26.89 and 20.92 percentage points, respectively. The proportions of DOC/SOC and MBC/SOC were all highest in CK, and the DOC/SOC and MBC/SOC in the soils of the remaining years were significantly lower relative to CK ($p < 0.05$). In the soil layer from 20 to 40 cm, the proportion of EOC/SOC was the highest in Y5, which significantly decreased by 11.02 and 9.24 percentage points when tea was planted for 20 and 40 years, respectively, compared with Y5 ($p < 0.05$). The proportion of POC/SOC was the highest in Y50, which significantly increased by 46.63 compared with CK ($p < 0.05$). The proportion of DOC/SOC was the highest in CK, which significantly increased by 46.63 compared with CK ($p < 0.05$). The proportion of MBC/SOC was the highest in CK, which significantly decreased with the years of tea planting, DOC/SOC ratio gradually decreased. MBC/SOC ratio was highest in Y5, which significantly increased by 1.56 percentage points ($p < 0.05$) compared to CK, however, with the increase of tea planting time, Y15, Y20, Y40, and Y50 significantly decreased by 1.48, 1.87, 1.96, 1.84 percentage points, respectively ($p < 0.05$).

**Table 2 Percentage of organic carbon fraction to organic carbon (%) for different tea planting years.**

| Soil layer | Years | EOC/SOC | POC/SOC | DOC/SOC | MBC/SOC |
|---|---|---|---|---|---|
| 0–20 cm | 0 | 21.53 ± 2.41abc | 44.42 ± 18.09c | 1.09 ± 0.26a | 0.54 ± 0.26a |
| | 5 | 23.91 ± 2.33a | 50.39 ± 9.78bc | 0.53 ± 0.03b | 0.27 ± 0.1bc |
| | 15 | 18.95 ± 2.25bc | 59.99 ± 5.56abc | 0.47 ± 0.09b | 0.31 ± 0.08b |
| | 20 | 17.22 ± 2.35c | 65.96 ± 8.32ab | 0.41 ± 0.1b | 0.18 ± 0.06bc |
| | 40 | 19.28 ± 2.27abc | 71.31 ± 2.27a | 0.37 ± 0.1b | 0.05 ± 0.02c |
| | 50 | 22.99 ± 4.22ab | 66.88 ± 8.25ab | 0.36 ± 0.14b | 0.22 ± 0.07bc |
| 20–40 cm | 0 | 22.46 ± 2.01ab | 21.46 ± 2.64d | 1.26 ± 0.07a | 0.55 ± 0.10b |
| | 5 | 26.00 ± 6.27a | 25.83 ± 10.83cd | 0.97 ± 0.04b | 2.11 ± 0.85a |
| | 15 | 19.41 ± 8.69ab | 42.09 ± 3.69bc | 0.62 ± 0.10cd | 0.63 ± 0.26b |
| | 20 | 14.98 ± 3.04b | 51.60 ± 22.27ab | 0.50 ± 0.13d | 0.24 ± 0.04b |
| | 40 | 16.76 ± 1.05b | 64.09 ± 10.30a | 0.80 ± 0.13bc | 0.15 ± 0.02b |
| | 50 | 18.73 ± 4.47ab | 68.09 ± 6.75a | 0.62 ± 0.18cd | 0.27 ± 0.06b |

**Note:**
Significant differences between different periods are indicated by different lowercase letters ($p < 0.05$)

## Soil bacterial alpha diversity with different tea planting duration

Alpha diversity includes Chao 1, ACE, Shannon and Simpson indices, *etc*. Chao 1 and ACE indices can reflect the abundance of soil microbial communities, and the higher its value, the richer the community species; Shannon and Simpson are used to estimate the microbial diversity characteristics, and the higher the value, the higher the community diversity and homogeneity. The changes in Alpha diversity of soil bacterial communities in tea gardens under different years of tea cultivation are shown in Table 3. With increasing years of tea cultivation, the richness of bacterial communities changes. Both the Chao 1 index (1,930~3,077) and ACE index (2,068~3,155) show the same pattern, with Y40 < Y50 < CK < Y20 < Y5 < Y15, indicating an increase-decrease-increase trend in bacterial community richness with increasing years of tea cultivation, reaching the highest richness at 15 years of tea cultivation. Compared to CK, the bacterial richness increased in Y5, Y15, and Y20, while it decreased in Y40 and Y50, with a significant decrease of 32.52% in richness at Y40 ($p < 0.05$). However, with the extension of tea cultivation time, the Chao index and ACE index of Y50 rose again to 2,477 and 2,598 respectively, indicating a gradual increase in bacterial community richness. From the Shannon and Simpson indexes, tea planting had a significant impact on the diversity change of the bacterial community structure of the tea garden soil, which also showed a trend of increasing and then decreasing, compared with CK, the diversity at 15 years of tea planting was the largest, with the Shannon and Simpson indexes reaching 6.69 and 0.99, respectively, and the diversity at 40 years of tea planting was the smallest. Shannon index and Simpson index decreased to 5.31 and 0.97, respectively.

**Table 3 Alpha diversity index of soil bacteria in different tea planting period.**

|  | Chao1 | ACE | Shannon | Simpson |
|---|---|---|---|---|
| CK | 2,860.83 ± 37.58ab | 2,945.53 ± 16.8a | 6.38 ± 0.08ab | 0.9951 ± 0.00a |
| Y5 | 2,946.88 ± 107.26ab | 2,997.13 ± 111.26a | 6.57 ± 0.03a | 0.9965 ± 0.00a |
| Y15 | 3,077.84 ± 116.75a | 3,155.22 ± 145.26a | 6.69 ± 0.07a | 0.9969 ± 0.00a |
| Y20 | 2,936.55 ± 58.53ab | 3,051.91 ± 74.77a | 6.52 ± 0.03a | 0.9957 ± 0.00a |
| Y40 | 1,930.53 ± 92.56c | 2,068.01 ± 97.73b | 5.31 ± 0.17c | 0.9781 ± 0.01b |
| Y50 | 2,477.25 ± 395.59bc | 2,598.08 ± 401.8ab | 5.76 ± 0.51bc | 0.9839 ± 0.01ab |

**Note:**
Significant differences between different periods are indicated by different lowercase letters for the same indicator ($p < 0.05$).

## Soil bacterial community composition with different tea planting duration

### Differences in soil bacterial communities among tea planting years

As shown in Fig. 3, the grouping of points in different colors represents soil samples from tea gardens of different cultivation years. If two group sample points are closer, it indicates a more similar species composition between the tea gardens. The results showed that the soil samples from tea gardens of all ages were more converged together and there was a certain distance between groups, indicating that there were some differences in bacterial species between the soil environments of tea gardens of different ages. Compared to CK, the points in the Y20, Y40, and Y50 groups are relatively distant, indicating differences in their community structures. Some overlap is observed between samples of CK, Y5, and Y15, indicating higher similarity in bacterial communities between them. Moreover, with stress <0.1, it suggests that NMDS analysis represents soil bacterial grouping samples well. Overall, there was some variability in the bacterial community structure, indicating that tea planting would change the soil bacterial community structure, which would converge within a certain period, while when the tea planting time reached a certain number of years, the bacterial community structure began to change.

### Soil bacterial community composition with different tea planting duration

As shown in Fig. 4, a total of 10 dominant bacterial phyla were obtained from the soil bacteria of tea gardens with different years of tea planting, which were *Acidobacteriota, Chloroflexi, Proteobacteria, Actinobacteriota, Bacteroidota, Verrucomicrobiota, WPS-2, Firmicutes, RCP2-54,* and *Gemmatimonadota,* with the highest abundance of the top 10 species in soil Y15 and the lowest in Y40. *Acidobacteriota* (21.77%~38.08%), *Chloroflexi* (16.79%~28.42%), *Proteobacteria* (14.60%~27.86%), and *Actinobacteriota* (3.70% ~10.39%) were the dominant phylum in the soil bacteria of the tea plantation. Their relative abundance sums could up to 84.04%, 86.00%, 71.64%, 76.43%, and 84.55% in the tea plantations from Y5 to Y50, respectively. Compared to CK, the relative abundance of *Acidobacteriota* gradually decreases with increasing years of tea cultivation, reaching its lowest abundance at Y40, which is 21.77%, a decrease of 16.32% compared to CK. Meanwhile, the relative abundance of *Chloroflexi* shows an increasing trend compared to the control group (CK), with the highest relative abundance at Y50, reaching 28.42%, an

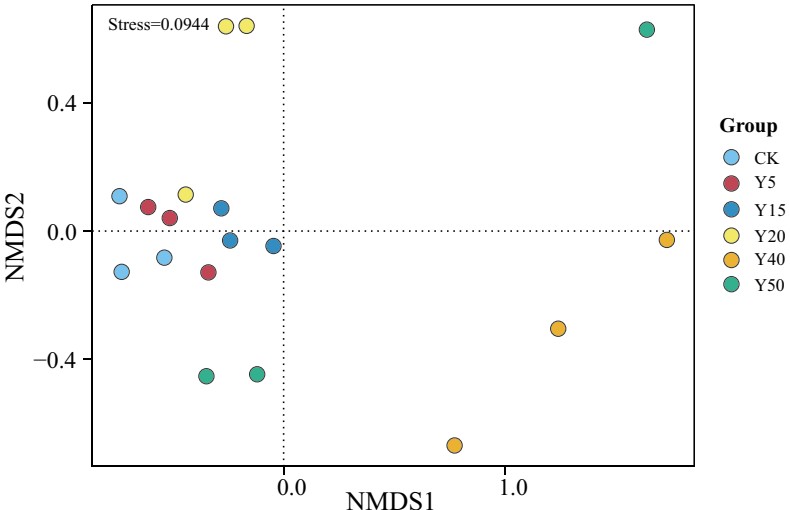

**Figure 3  NMDS analysis of soil bacterial communities in different tea planting period.**

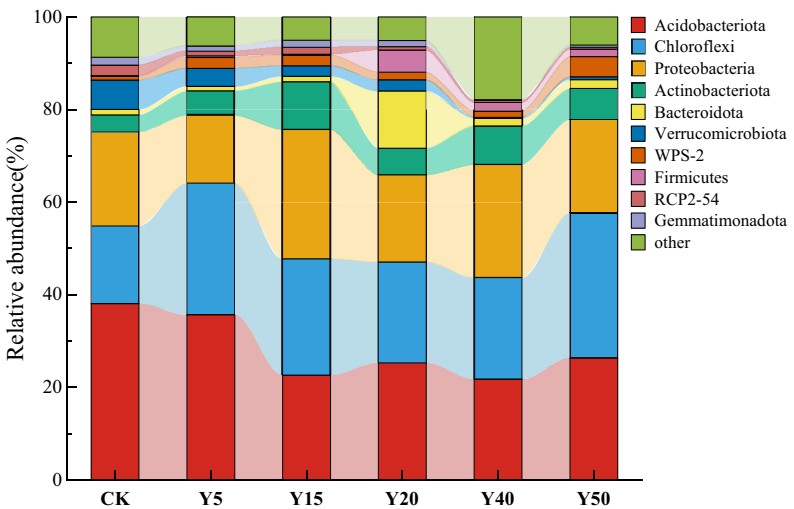

**Figure 4  The bacterial composition and relative abundance of communities at the phylum.**

increase of 11.64% compared to CK. Additionally, the relative abundance of *Chloroflexi* at Y20 and Y40 is relatively lower compared to Y5 and Y15. *Proteobacteria* and *Actinobacteriota* exhibit similar trends in relative abundance, reaching their maximum abundance at Y15 compared to CK, gradually decreasing afterward, and then increasing again.

### *Prediction of soil bacterial function with different tea planting duration*

Based on FAPROTAX, a study was conducted on bacterial functional groups in soil samples from different years of tea cultivation, resulting in the annotation of 55 functional types. Among them, the top ten relatively abundant bacterial functions were selected for plotting, as shown in Fig. 5. The main functional groups of soil bacteria include *aerobic*

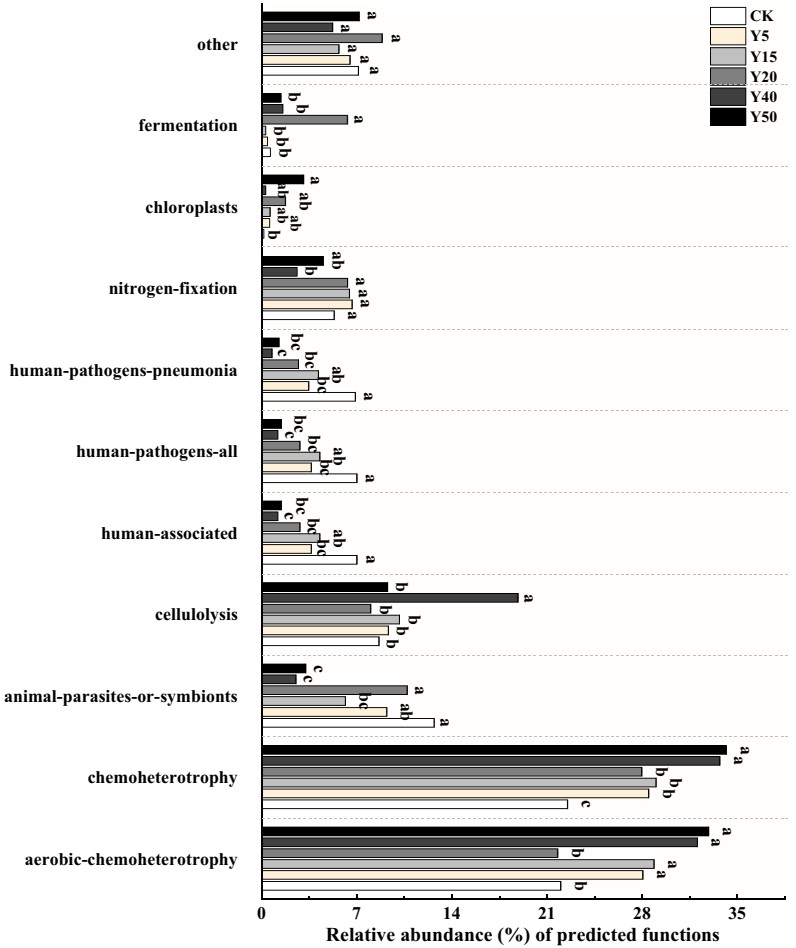

**Figure 5 Relative abundance of bacterial functions based on FAPROTAX prediction (TOP10).**
Significant differences between different periods are indicated by different lowercase letters
($p < 0.05$).            

*chemoheterotrophy* (21.76% to 32.88%), *chemoheterotrophy* (22.50% to 34.18%), *cellulolysis*
(8.02% to 18.84%) and *nitrogen fixation* (2.62% to 6.65%). Functional categories involved
in soil carbon and nitrogen transformation include *aerobic chemoheterotrophy*, *cellulolysis*,
*nitrogen fixation*, and *chloroplast*. There were differences in soil bacterial functions
between tea planting years. After tea planting, *aerobic chemoheterotrophy* functional
bacteria increased significantly, and the relative abundance was significantly higher
($p < 0.05$) than CK in all tea planting years except Y20. The proportional abundance of
*cellulolysis* functional bacteria was highest in Y40, which differed significantly ($p < 0.05$)
compared to the other tea planting years of soil. The relative abundance of *nitrogen
fixation* bacteria was significantly higher ($p < 0.05$) in soils of 5, 15, 20, and 50 years of tea
planting than in Y40 and also higher than in CK, while there was no significant difference.
The proportional abundance of *chloroplast* bacteria was higher ($p < 0.05$) in soils of Y20
and Y50 than in those of Y5, Y15, and Y40, and was obviously higher ($p < 0.05$) in Y50
than in CK.

### Soil organic carbon and microbial community correlation

*The correlation between soil organic carbon and dominant taxa in bacterial communities*

The effect of SOC on different bacterial populations under tea planting years was explored based on redundancy analysis with SOC content and its fractions as explanatory variables and the top 10 species in relative abundance at the soil bacterial phylum level as response variables. The results, as shown in Fig. 6, showed that the first and second ordering axes explained a total of 34.14% of the overall changes in SOC and its components, with POC being able to significantly influence the community composition of bacteria (F = 3.4, $P = 0.002$), explaining 17.6% of the changes with a 42.5% contribution, followed by MBC, which was able to explain 9.5% of the changes, with a contribution of 23.0% (Table 4). The angles between the arrows of SOC, EOC, and POC and the arrows of *Verrucomicrobiota*, *Gemmatimonadota*, and *Acidobacteriota* were small and in the same direction, indicating a strong positive correlation. However, there was a negative correlation with *Bacteroidota*, *Proteobacteria*, *Chloroflexi*, and *Actinobacteriota*. MBC had a strong positive correlation mainly with *Proteobacteria* and *RCP2-54*, and a negative correlation with *Chloroflexi* and *Actinobacteriota*. DOC had a strong positive correlation with *Acidobacteriota* and a negative correlation with *Firmicutes* and *Actinobacteriota*.

*The correlation between soil organic carbon and functional groups of bacteria*

Redundancy analysis of soil organic carbon (SOC) and dominant bacterial functional taxa was performed, and the results are shown in Fig. 7, where the first and second ordering axes explained a total of 49.57% of the overall changes in SOC and its components on bacterial functional taxa. Soil POC and MBC had a significant effect on bacterial functional groups (Table 5), with soil particulate organic carbon independently explaining 24.3% of the variation, with a contribution of 46.7% (F = 5.1, $P = 0.008$), and microbial biomass carbon independently explaining 16.3% of the variation, with a contribution of 31.6% (F = 4.1, $P = 0.018$). From the correlation of organic carbon and each carbon fraction with the main functional groups of the measured soil bacteria, there was a significant positive correlation between MBC and nitrogen fixation, a strong negative correlation between POC, SOC, and EOC mainly with chloroplast nitrogen fixation and chemo-energetic heterotrophs, while cellulose hydrolysis had a negative correlation mainly between MBC and DOC.

## DISCUSSION

### Effect of tea planting on soil organic carbon and its fractions in tea plantations

Organic carbon is critical for soil nutrient cycling, aggregate formation, water retention, and biodiversity, and it directly affects the carbon sink potential and fertility level of soils and is often used to assess soil quality (*Bongiorno et al., 2019*; *Bünemann et al., 2018*). In this study, during the 0–50 years of tea planting, the SOC content was mainly distributed in the 0–20 cm soil layer, and tea cultivation promoted the accumulation of soil SOC content

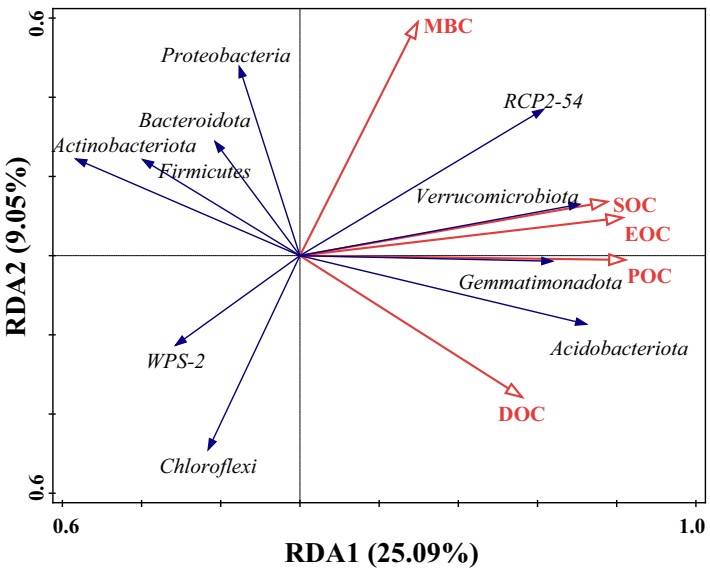

**Figure 6 Redundancy analysis (RDA) of SOC and bacteria dominant groups.** Hollow red arrows indicate SOC and fractions, and solid blue arrows indicate dominant groups, the same below.

**Table 4 Interpretation of soil environmental factors on changes of bacterial community components.**

| Name | Explains% | Contribution% | pseudo-F | P |
|------|-----------|---------------|----------|------|
| POC | 17.6 | 42.5 | 3.4 | 0.006 |
| MBC | 9.5 | 23 | 2 | 0.068 |
| SOC | 6.1 | 14.8 | 1.3 | 0.264 |
| EOC | 4.3 | 10.4 | 0.9 | 0.504 |
| DOC | 3.9 | 9.4 | 0.8 | 0.526 |

in the tea plantation, and the soil organic carbon content of the tea plantation after tea planting was higher than that of CK, which began to change significantly at 20 years, and there was a maximum content at 40 years, and then there was a decreasing change at 50 years. This trend was similar to the results of previous studies (*Kong, 2016*; *Di et al., 2020*). The main reasons for the increase in organic carbon content may be the litter of tea trees, root exudates, and nitrogen fertilizer application (*Li et al., 2015*). Plant litter directly inputs organic matter into the soil (*Spohn, Babka & Giani, 2013*), while root exudates, characterized by low molecular weight and high bioavailability, are easily converted by microorganisms into mineral carbon, thereby increasing the organic carbon content (*Liang, Schimel & Jastrow, 2017*; *Yang et al., 2023*). Additionally, nitrogen fertilizer application promotes the formation of alkyl carbon and aromatic carbon, reducing the content of alkoxy carbon. Alkoxy carbon is the most easily degradable component of SOC, with a less stable structure. Therefore, nitrogen application promotes the stability and sequestration of soil organic carbon (*Li et al., 2019a*). Furthermore, the predominant

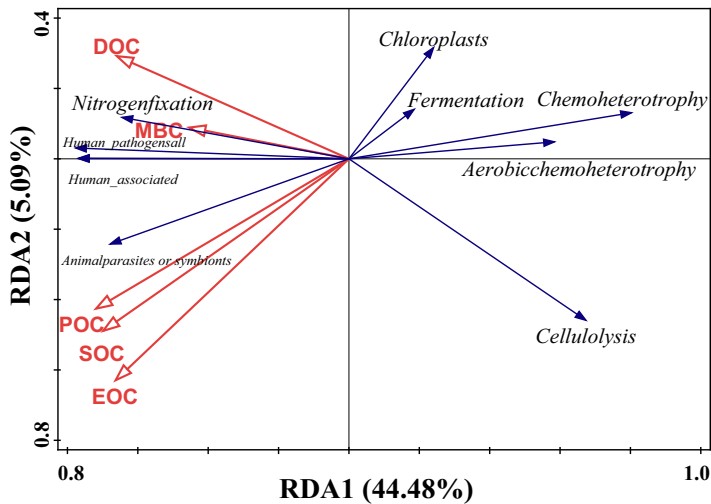

**Figure 7 Redundancy analysis of SOC and relative abundance of bacterial functions.**

**Table 5 Interpretation of soil environmental factors on changes of bacterial community function.**

| Name | Explains% | Contribution% | pseudo-F | P |
|---|---|---|---|---|
| POC | 24.2 | 46.7 | 5.1 | 0.008 |
| MBC | 16.3 | 31.6 | 4.1 | 0.018 |
| DOC | 7.3 | 14.1 | 2 | 0.122 |
| EOC | 3 | 5.8 | 0.8 | 0.506 |
| SOC | 0.9 | 1.8 | 0.2 | 0.908 |

climatic type is a meso-subtropical monsoon humid climate, characterized by moderate temperatures and an average annual precipitation of 1,375 millimeters within the study area. Therefore, in comparison to the high rainfall volumes of tropical nations, the region's lesser erosive power likely constitutes a favorable condition for the accumulation and preservation of organic carbon (*Berhe et al., 2008*; *Liu et al., 2023*). In this study, it was found that the organic carbon content of the soil significantly increased after 20 years. This increase could be attributed to the high microbial activity during this stage, leading to a higher decomposition and utilization rate of organic matter. A large amount of active substances were decomposed under the action of microorganisms, resulting in a stronger degree of soil humification (*Wang, Li & Zheng, 2017*). However, after 50 years of tea plantation, the decrease in organic carbon content may be due to the decrease in organic matter secretion by tea roots after long-term tea planting, thereby reducing the accumulation of organic carbon in older tea gardens (*He, Zheng & Zhu, 2021*).

The active organic carbon fraction is characterized by a short turnover time and fast rate and is more sensitive to changes in the external environment conditions (*Xu et al., 2021*). Therefore, it is beneficial to study the changes of active organic carbon components in the soil of tea plantations with different years of tea planting to have a more comprehensive
understanding of the dynamics and turnover of soil carbon in the context of tea planting. In this study, it was found that POC and EOC play a predominant role in active organic carbon. Compared to CK, after tea planting, the content of POC and EOC was higher, and the longer the tea plantation time, the more accumulation occurred. Existing studies have indicated that POC is considered persistent soil organic carbon, associated with plants. After plant litter is input into the soil, POC is enclosed within aggregates and forms organic-mineral complexes (*Witzgall et al., 2021*). EOC has a certain degree of leaching, which is easily mineralized and utilized by microorganisms. The terrain of the study area is hilly with some slopes, whereas with the increase of tea tree cover, organic litter, and root system, rainfall is intercepted by the tea tree, and EOC loss is reduced (*Zhang et al., 2023*), whereas the enhancement in microbial activity promotes the formation of EOC (*Zhao et al., 2023*). DOC is mainly derived from short-term dead leaves and humus in soil organic matter, which is susceptible to soil leaching (*Guggenberger & Kaiser, 2003*). In this study, we found that the DOC content was significantly lower at 5 years of tea planting compared with CK, which may be because the tea trees were just planted at that stage, with low cover, few dead branches and leaves, and the bare surface was unfavorable for organic carbon conservation (*Hu et al., 2023*). However, the canopy becomes thicker, and the surface layer of dead and leafy material increases with years of tea planting, which protects DOC (*Zhu et al., 2018*). This study found that at 40 years of tea plantation, the MBC content was the lowest, significantly lower than other tea plantation durations, but it gradually recovered by 50 years. We speculated that the sharp decline in soil pH caused by tea plantations could be the reason (*Yan et al., 2020b*). As shown in Table 1, the soil pH gradually decreased from 0 to 50 years after the tea plantation, leading to problems such as aluminum toxicity (*Dong et al., 1999*), which affected the living environment of soil microorganisms and reduced microbial population. However, after 50 years of tea plantation, soil microorganisms gradually adapted to this acidic environment, regained activity, and facilitated microbial carbon transformation. In addition, our study also found that the accumulation of organic carbon and its fractions in the lower soils was slower than in the surface soils. This may be attributed to the more litter material and stronger microbial activity in the 0–20 cm layer, whereas the reduced presence of plant roots in the 20–40 cm layer leads to lower organic carbon and component content (*Prescott & Vesterdal, 2021*).

The distribution ratio of active organic carbon components better reflects the cycling and transformation of soil carbon pools, indicating the availability of soil organic carbon and the smaller the ratio is, the more the soil lacks easy-to-use carbon sources (*Liao & Boutton, 2008*), which is conducive to the accumulation of organic carbon. From Table 3. 2-1, it is observed that with the extension of tea plantation time, the ratios of EOC/SOC, DOC/SOC, and MBC/SOC generally decrease, while the ratio of POC/SOC gradually increases. This may be because POC mainly originates from undecomposed or partially decomposed plant residues (*Dijkstra, Zhu & Cheng, 2021*), primarily composed of cellulose, hemicellulose, and plant-derived phenolic substances (such as lignin phenols), which have characteristics of low density, high C/N ratio, and susceptibility to decomposition, but are often underutilized by plants due to their complex chemical composition and high activation energy (*Jilling et al., 2018*; *Williams et al., 2018*).

Therefore, although planting tea trees promotes the accumulation of organic carbon and its active components, the presence of POC, which is not favorable for direct absorption and utilization by tea trees, suggests that tea plantation may be more conducive to the formation of particulate organic carbon, thus facilitating organic carbon transformation.

## Effect of tea planting on soil bacteria communites

The richness and diversity of soil bacterial communities are important indicators of soil quality changes (*Mhete et al., 2020*). This study found that with the increase in the years of tea plantation, both bacterial richness index and diversity index showed an increasing trend initially and subsequently diminishing. The Chao index, ACE index, Shannon index, and Simpson index of Y15 had been appreciably higher, indicating that around 15 years of tea plantation, the richness and diversity of microbial communities reached their peak. This is consistent with previous research findings (*Xu et al., 2020*). Because tea planting provides more space and material for microbial growth, which favors the growth and multiplication of soil bacteria (*Han, Kemmitt & Brookes, 2007*). From 15 years, soil bacterial richness and diversity began to decline and when the tea planting reached 40 years, the bacterial community was significantly reduced, probably because the specialized habitats formed by long-term tea planting inhibited the development of soil bacterial diversity, *e.g.*, by the significant decrease in soil pH (as shown in Table 1) (*Cao, Liu & Li, 2011*; *Ye et al., 2016*; *Wan et al., 2020*). On the other hand, it could also be the reason for continuous monocropping in the same area, which has been reported in previous studies to lead to a decrease in crop yield and quality, as well as to enrichment of pathogenic bacteria and the reduction of beneficial microorganisms (*Wang et al., 2023a*).

Statistical analyses showed that the effect of tea planting on the structure of soil bacterial communities in tea plantations produced differences, and *Acidobacteriota, Chloroflexi, Proteobacteria, Actinobacteriota* were the dominant phyla in the soil bacteria of the tea plantations, which is in agreement with the results of the previous studies (*Wu et al., 2023*). In this study, the relative abundance of *Proteobacteria* showed a 'V'-shaped trend with increasing years of tea plantation, while nitrogen-fixing rhizobia, which belongs to the *Proteobacteria*, is important functional bacteria (*Jones et al., 2009*). Therefore, we speculated that the change in the relative abundance of Proteobacteria may be related to nitrogen fixation in the soil. From the point of view of bacterial functional groups, the change of nitrogen fixation may also change with the change of Aspergillus phylum, which is strong and then weak with the increase of tea planting years, and its relative abundance reaches the maximum in 15 years, and then shows a V-shaped change in the period of tea planting from 20 to 40 years. *Chloroflexi* and *Acidobacteriota* are soil oligotrophic bacteria that grow slowly, often utilize difficult-to-decompose and nutrient-poor substrates for metabolism, and dominate in soils with low organic carbon quality and quantity (*Fierer, Bradford & Jackson, 2007*). Therefore, their relative abundance in this study was higher in unplanted tea, 5 and 15 years of tea planting. Meanwhile, the *Chloroflexi*, which includes photooxygenated bacteria, is also able to cultivate strains that achieve aerobic heterotrophy (*Hanada, 2014*; *Islam et al., 2019*). Our study suggests that the nitrogen fixation by

chloroplasts and the community of aerobic chemoautotrophic bacteria after tea planting may mainly come from the phylum *Chloroflexi*.

However, in the results of this study, the relative abundance of *Acidobacteriota* showed a decreasing trend in tea plantation soils aged 20 to 50 years, which differs from the research findings of *Wang, Tang & Ye (2021)*. This difference may be connected with the increase in organic carbon and active component content. *Guo et al. (2021)* found that higher carbon availability can inhibit its relative abundance. As shown in Fig. 4, the relative abundance of *Acidobacteriota* after tea planting was higher than in unplanted soil, indicating that tea planting promotes the growth of *Acidobacteriota* in the soil bacterial community. Most microorganisms in *Acidobacteriota* can metabolize to produce antibiotics, plant hormones, hydrolytic enzymes, and other active ingredients. These substances enhance the plant's physiological and biochemical characteristics against adversity, playing important roles in plant growth, microbial community improvement, and soil development (*De Simeis & Serra, 2021*; *Olanrewaju & Babalola, 2019*). Therefore, *Acidobacteriota* are more adaptable to environmental changes than other microorganisms. They can isolate acidophilic actinomycetes from acidic soils (*Zakalyukina Yu, Zenova & Zvyagintsev, 2002*; *Poomthongdee, Duangmal & Pathom-aree, 2015*), which may also be a reason for the increase in their abundance after a significant decrease in tea planting for 20 years. In summary, in the process of tea planting, we recommend intercropping other crops between the rows of tea trees to avoid the monoculture affecting the microbial activity of the bacterial community. At the same time, in the process of tea planting, take corresponding measures to prevent soil acidification, ensure the maintenance of a good ecological environment of the tea plantation soil, and achieve the sustainable development of the tea industry.

## Effect of soil organic carbon fractions on bacterial communities in tea plantations

Organic matter decomposition enhances the soil microbial community's association with nutrient cycling (*Manasa et al., 2020*). Our study found that POC and MBC were the primary factors influencing the bacterial community. This study is consistent with previous research findings (*Zhang et al., 2021a*; *Yan et al., 2020a*). Previous studies have indicated that during the decomposition of tea tree pruning materials or litter, early-stage water-soluble substances are easily leached out, and labile substances degrade rapidly. However, as decomposition progresses, recalcitrant substances such as cellulose and lignin accumulate, slowing down the decomposition rate (*Berg & McClaugherty, 2020*). Therefore, as the duration of tea planting increases, such substances accumulate more in tea plantation soils, becoming the primary source of particulate organic carbon (*Dijkstra, Zhu & Cheng, 2021*), requiring more microbial activity to aid in their decomposition and transformation. Therefore, the present study inferred that with the increase in the number of years of tea planting, the changes in the quality and quantity of litters material made the soil bacterial community may have shown differences with the changes in POC content. Our study found that the functional groups of soil bacterial communities in tea plantation soils were influenced by MBC, likely affected by the decrease in pH after tea planting. MBC

directly controls soil extracellular enzyme activity, which is the basis for driving SOC turnover (*Joergensen & Wichern, 2018*; *Li et al., 2021*). Whereas, soil pH is an important factor limiting soil microbial activity (*Li et al., 2010*), which influences the production of biological enzyme activities. In this study, we observed a consistent relationship between the variation in the relative abundance of *Proteobacteria* and changes in MBC. This consistency may stem from the ability of *Proteobacteria* to utilize nutrients such as ammonia and methane produced during the decomposition of organic matter for growth and metabolism (*Lv et al., 2014*), thus participating in soil carbon cycling. On the other hand, it may be due to the associated flora in the phylum Ascomycota with nitrogen-fixing function (*Jones et al., 2009*), so that MBC showed a positive correlation with nitrogen fixation. In addition, there was a negative correlation between MBC and cellulolysis, which may be related to the increase in POC content as well as to the *Actinobacteriota*, which declined at 40 years of tea planting, whereas *Thermoanaerobacteria* of the *Actinobacteriota* can produce cellulose-degrading enzymes for direct degradation of cellulose from litters material (*Mitra et al., 2022*), which participates in the decomposition and synthesis of humic substances (*Bhatti, Haq & Bhat, 2017*), thereby promoting the conversion of POC and inhibiting the formation of MBC. In conclusion, this also implies that *Proteobacteria* and *Actinobacteriota* may be more susceptible to environmental changes but can balance this variability well. Therefore, after 50 years of tea planting, their abundance begins to increase, continuing to contribute to soil carbon cycling.

## CONCLUSIONS

Cultivating tea trees promotes the accumulation of soil organic carbon and its active components, a crucial factor in maintaining the high-quality development of the tea industry. Research findings indicate a significant enhancement in soil organic carbon and its active components within tea gardens over 50 years of tea planting. Specifically, soil organic carbon, particulate organic carbon, and dissolved organic carbon experienced substantial increases after 20 years of tea cultivation, likely due to contributions from tea litter, root exudates, and nitrogen fertilizer application. By the 15 years of tea planting, there was a notable increase in the richness and diversity of soil bacterial communities, potentially attributed to tea planting providing increased growth space and resources for microbial proliferation, thereby facilitating soil bacteria growth and reproduction. However, this richness experienced a significant decrease by the 40 years of tea planting. Soil POC and microbial biomass carbon were identified as significant factors influencing soil bacterial communities, potentially impacted indirectly by changes in soil pH resulting from tea cultivation. Therefore, while research indicates the benefits of planting tea trees for enhancing soil organic carbon and its components, it is advisable to intercrop other crops between tea rows during planting to mitigate the adverse effects of monoculture on bacterial community activity. Additionally, measures should be implemented to prevent and manage soil acidification during tea cultivation, ensuring the preservation of a favorable ecological environment in tea gardens and fostering sustainable development in the tea industry.

### Funding

This work was supported by the Talent Introduction Fund of Guizhou University-Guizhou University Key Hopes [2017] No. 47, the Guizhou Provincial Science and Technology Programme Project-Guizhou Keheji ZK [2022] Key 014, the Science and Technology Programme Project of Guiyang Municipality-Zhu Ke Contract [2024]-1-13. The funders had no role in study design, data collection and analysis, decision to publish, or preparation of the manuscript.

### Grant Disclosures

The following grant information was disclosed by the authors:
Talent Introduction Fund of Guizhou University-Guizhou University Key Hopes [2017] No. 47.
Guizhou Provincial Science and Technology Programme Project-Guizhou Keheji ZK [2022] Key 014.
Guiyang Municipality-Zhu Ke Contract [2024]-1-13.

### Competing Interests

The authors declare that they have no competing interests.

### Author Contributions

- Yingge Shu conceived and designed the experiments, performed the experiments, analyzed the data, prepared figures and/or tables, authored or reviewed drafts of the article, and approved the final draft.
- Shan Xie conceived and designed the experiments, performed the experiments, analyzed the data, prepared figures and/or tables, authored or reviewed drafts of the article, and approved the final draft.
- Hong Fan conceived and designed the experiments, performed the experiments, authored or reviewed drafts of the article, and approved the final draft.
- Chun Duan conceived and designed the experiments, performed the experiments, authored or reviewed drafts of the article, and approved the final draft.
- Yuansheng Liu conceived and designed the experiments, performed the experiments, authored or reviewed drafts of the article, and approved the final draft.
- Zuyong Chen conceived and designed the experiments, performed the experiments, analyzed the data, prepared figures and/or tables, authored or reviewed drafts of the article, and approved the final draft.

### DNA Deposition

The following information was supplied regarding the deposition of DNA sequences:
The tea plantation soil sample raw sequence reads described here are accessible *via* BioProject (NCBI's SRA database) accession number PRJNA1104766.

## Data Availability

The raw data shows tea planting improves organic carbon accumulation.

## Supplemental Information

Supplemental information for this article can be found online at http://dx.doi.org/10.7717/peerj.18683#supplemental-information.

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
