# Peer review of "Tea cultivation: facilitating soil organic carbon accumulation and altering soil bacterial community—Leishan County, Guizhou Province, Southwest China"

_PeerJ, doi:10.7717/peerj.18683_

## Round 0.1 · original submission · Minor Revisions

This article has been extensively reviewed and all 3 reviewers have only requested Minor Revisions. Please address all review comments and resubmit

·

Basic reporting

The English language is fine. I did not detect issues. However, some sentences are overly long and redundant. I suggest that the authors rewrite these sentences for better readability and conciseness. Remove the extra punctuation like commas used several times in a single sentence.
The literature references cited in the manuscript are sufficient, appropriate, and relevant to the research objectives. The article structure, figures, and tables meet the criteria of PeerJ. The study hypothesis is described as scientifically sound. Therefore, I don't find any reporting basic issues in this submission.

Experimental design

The manuscript presents original primary research that aligns well with the aims and scope of the journal. The research question is well-defined, relevant, and meaningful, addressing an identified knowledge gap. The investigation is conducted with rigor, adhering to high technical and ethical standards. The methods are described in sufficient detail to enable replication.

Validity of the findings

The amnuscript findings are valid. The manuscript demonstrates a strong adherence to rigorous scientific standards. All underlying data have been provided, and they are robust, statistically sound, and appropriately controlled. The conclusions are well stated and effectively linked to the original research question, limited to supporting results. Therefore, I don't have an issue with the validity of the study findings.

Additional comments

The manuscript meets the criteria for publication in terms of relevance, methodological rigor, and clarity. I recommend that the manuscript be accepted for publication after incorporating and answering the comments I made in the original file. I gave different colors to the comments in the original draft, please answer accordingly.

·

Basic reporting

Comments are digitally given in the manuscript

Experimental design

Comments are digitally given in the manuscript particularly where elaborations are necessary and also the clear statements (e.g., abbreviation CK )

Validity of the findings

Comments are digitally given in the manuscript particularly where findings are concerned for the readers to understand better

Additional comments

Author/s must respond to comments as appropriate

Reviewer 3 ·

Basic reporting

No comment

Experimental design

I suggest author to add effect of organic litters (shed leaves, mulch materials, shade tree leaves etc.) on soil orgqnic matter content and microbial characteristics at least mention these factors in the discussion and introduction.

Validity of the findings

No comment

Additional comments

A worldwide perspective is somewhat missing. Authors may add reports of similar works carried out in other tea growing regions of the world and their significance. Effect of ta garden biomass weed biomass on soil organic carbon, soil microbes etc. can be discussed with references if any. However, I accept this article for publication with these minor revision.

---

## Round 0.2 · accepted · Accept

Congratulations.
Yours,
Yoshi
Prof. Yoshinori Marunaka, M.D., Ph.D.

·

Basic reporting

no comment

Experimental design

no comment

Validity of the findings

no comment

·

Basic reporting

No comment, as the points raised and clarifications sought in the review of MS at its first read have been attended to and satisfactory responses have been provided for the clarifications.

Experimental design

No comments, as explanations to 1st review of MS recommendations have been given and incorporated to revised MS duly.

Validity of the findings

No comment, findings are clearer.

Additional comments

Revised MS now reads in order.
I would like to receive a published MS for educational purpose and only for reference